# Bat Pass Duration Measurement: An Indirect Measure of Distance of Detection

**Christian Kerbiriou** [1,2,*][ID]**, Yves Bas** [1,3]**, Isabelle Le Viol** [1,2]**, Romain Lorrillière** [1][ID]**, Justine Mougnot** [1] **and Jean-François Julien** [1][ID]

[1] Centre d'Ecologie et des Sciences de la Conservation (CESCO), Muséum national d'Histoire naturelle, Centre National de la Recherche Scientifique, Sorbonne Université, CP 135, 57 rue Cuvier, 75005 Paris, France; yves.bas@mnhn.fr (Y.B.); isabelle.le-viol@mnhn.fr (I.L.V.); romain.lorrilliere@mnhn.fr (R.L.); justine.mougnot@gmail.com (J.M.); jfjulien@mnhn.fr (J.-F.J.)

[2] Centre d'Ecologie et des Sciences de la Conservation (CESCO), Muséum national d'Histoire naturelle, Station Marine, 1 place de la Croix, 29900 Concarneau, France

[3] Centre d'Ecologie Fonctionnelle et Evolutive (CEFE), Université de Montpellier—Université Paul-Valéry Montpellier—EPHE, route de Mende, 34293 Montpellier, France

* Correspondence: christian.kerbiriou@mnhn.fr

**Abstract:** Few reports have been published on detection distances of bat calls because the evaluation of detection distance is complicated. Several of the approaches used to measure detection distances are based on the researcher's experience and judgment. More recently, multiple microphones have been used to model flight path. In this study, the validity of a low-cost and simple detectability metric was tested. We hypothesize that the duration of an echolocating-bat-pass within the area of an ultrasonic bat detector is correlated with the distance of detection. Two independent datasets from a large-scale acoustic bat survey—a total of 25,786 bat-passes from 20 taxa (18 species and two genera)—were measured. We found a strong relationship between these measures of bat-pass duration and published detection distances. The advantages of bat-pass duration measures are that, for each study, experimenters easily produce their own proxy for the distance of detection. This indirect measure of the distance of detection could be mobilized to monitor the loss in microphone sensitivity used to monitor long-term population trends. Finally, the possibility of producing an index for distance of detection provides a weight for each bat species' activity when they are aggregated to produce a bat community metric, such as the widely used "total activity".

**Keywords:** bat activity; bat detectability; bat echolocation; bat-pass duration

---

## 1. Introduction

In the 1970s, the general opinion among bioacousticians was that bat species identification from echolocation signals was difficult [1]. Since then, knowledge and methods of acoustic identification of bat species have matured [2–4]. The cost of ultrasonic bat detectors and recorders has decreased, resulting in the development of passive acoustic sensors that are able to record throughout the night. To respond to this large number of records, several reliable quantitative methods for detecting sound events, extracting numerous acoustic features, and automatically identifying bat species have been developed [5–8]. Since the late 1990s, this non-intrusive method has been widely used by researchers to investigate habitat use by bats [9,10] or to evaluate the impact of various anthropogenic pressures, such as (1) agriculture [11,12]; (2) forestry [13,14]; (3) habitat fragmentation [15]; (4) non-lethal impacts of wind turbines, such as the disturbance of commuting and migration routes, local habitat loss [16–18], and ambient noise [19]; or (5) artificial light at night [20,21].

Among these studies, the "bat activity" measure is commonly based on the number of bat passes detected by a sensor per unit time. The metrics used for assessing bat activity vary among studies. Tibbels and Kurta used the number of pulses [13]; Hayes used the number of files recorded by bat detectors that include echolocation calls [22]. In acoustic studies using time expansion bat detectors, a bat-pass is defined as one or more bat echolocation calls during a sound recording. In this case, the duration of the record is predefined by an ultrasound detector [23–25]. Other studies calculated bat activity as the number of bat-passes per night, and a bat-pass was defined as a single bat call or several bat calls emitted during a fixed interval (5 s) [12,18,26]. Parsons and Jones also used the number of bat-passes per night to assess bat activity, but identified a bat-pass as a call sequence containing three or more pulses and, when the time between calls exceeded four times, the inter-pulse interval [5]. However, because it is impossible to know the exact number of individuals present when measurements are recorded, the resulting metric is only an index of activity. When a bat activity metric is used to examine the influence of the environment (such as habitat quality and anthropogenic pressure) on a single species, the implicit hypothesis is that the detectability is not influenced by the other factors, such as managed vs. unmanaged habitats or conventional vs. organic farming. This probability of detection varies among species according to the average sound pressure level of their call (dB), the frequency of the call, and the directionality of the emitted sound. For example, the beam pattern of an echolocation train is narrow and points forward in some species, such as *Rhinolophus ferrumequinum* [27], whereas the beam pattern is much less directional in species such as *Pipistrellus* spp. or *Myotis* spp. [28,29]. The detection distance depends not only on the specificity of the transmitter (i.e., bats), including sound amplitude and the flight pattern speed, sinuosity, and altitude, but also on the receiver (detector type and degradation over time), the medium (air temperature and humidity), the methods used to survey the bat activity (line transects and stationary detectors), and the effect of vegetation on sound attenuation [30,31].

Currently, few estimates of detection distances in bats have been published. One challenge with the bats is, in comparison with other taxa such as diurnal birds, the difficulties in intuitively estimating distances. At the time of writing, we only know of the distances of detection published for *Eptesicus bottae* [32], *Myotis lucifugus*, *Myotis leibii*, *Myotis septentrionalis*, and *Pipistrellus subflavus* [33]; for 11 species from Swaziland [34]; and for 27 French species [3] (Table S1). Forbes and Newhook [33] and Holderied et al. [32] performed their studies in a laboratory setting. The latter used the stereo videogrammetry method. Surlykke [35] found substantial differences between signals recorded in the laboratory and the field (higher directionality and intensity), emphasizing the value of studying animals in their natural habitat. However, the measurement of detection distance in the field is challenging. Monadjem et al. [34] recorded hand-released bats at different distances (i.e., when the bat commenced flying, observers immediately turned on the detectors to accurately assess the distance at which each bat was recorded). However, calls from hand-released bats may not be fully representative of free flying individuals [36]. Without knowing the specific details of the methodology used by Barataud [3] to assess those distances, we assumed that the method consisted of the capturing and marking of bats using a chemiluminescent tag and visually estimating the distance from observer to bat in flight at night [37–39]. However, this approach is partially based on the experience and judgment of the researcher (i.e., a visual estimation of the distances from animals in flight at night). Barataud [3] provided no details about the microphone used or habitats; thus, these empirical measures should be regarded rather as relative distances of detection. A more accurate assessment of detection distance could be achieved using recordings from an array of three or more spatially dispersed microphones and localization algorithms to determine the absolute geographic position of a sound source [40] and then to model the flight path. Until recently, this method has rarely been implemented in the field [40–42].

Accurate distance of detection is a key parameter for defining the region of detectability of a bat [34]. Estimating the volume of airspace sampled is an ongoing issue because comparisons of bat activity between sites could be biased by the environment (e.g., habitats), whereas such comparisons

could be needed when conducing pre-construction acoustic surveys at potential wind energy facility sites [43] or when analyzing bat mortality risks on roads [44]. Assessing differences in detectability across bat species allows a more accurate activity measure, particularly when species are pooled in the same index, such as the widely used "total activity" or other community metrics used in studies of bat assemblages [34,45]. Without any correction of this index, the abundances of species that are less detectable are underweighted. Finally, whether microphone degradation with use might introduce detection biases that would affect long-term monitoring and research projects must be determined [46].

Here, we tested the validity of a distance-detectability metric that is inexpensive, fast, and simple for use by bat workers. We hypothesized that the measure of bat-pass duration (i.e., each event expressed in seconds) of a bat detected within the area of a bat detector (Figure 1) (1) varies among species and (2) is correlated with the distance of detection. With the aim of testing this approach, two independent datasets were used (two recording protocols: along line transects and at stationary recorders) from the French bat monitoring program, a large-scale acoustic bat survey using two types of detectors operated by hundreds of volunteers since 2006. We tested the correlation between the distance of detection published by Barataud [3] and the average bat-pass duration using the data set from the French bat monitoring program. To evaluate the significance of bat-pass duration compared to alternative distance-detectability metrics, similar correlations were performed using two other widely used parameters of bat echolocation calls: frequency of peak energy and call duration. These two parameters are indirectly related to call intensity and, in turn, potentially to distance of detection [47]. For many echolocating bats, the peak frequency has been shown to be negatively correlated to body size [41,48]. Larger bats produce lower frequency sounds because they have a bigger larynx and larger resonant chambers [49]. Yet, calls of larger species are more intense [41] and thus have an expected greater distance of detection (Figure 2). Call duration is positively correlated with call intensity [47], so species with long pulsation duration are expected to have a greater distance of detection (Figure 2).

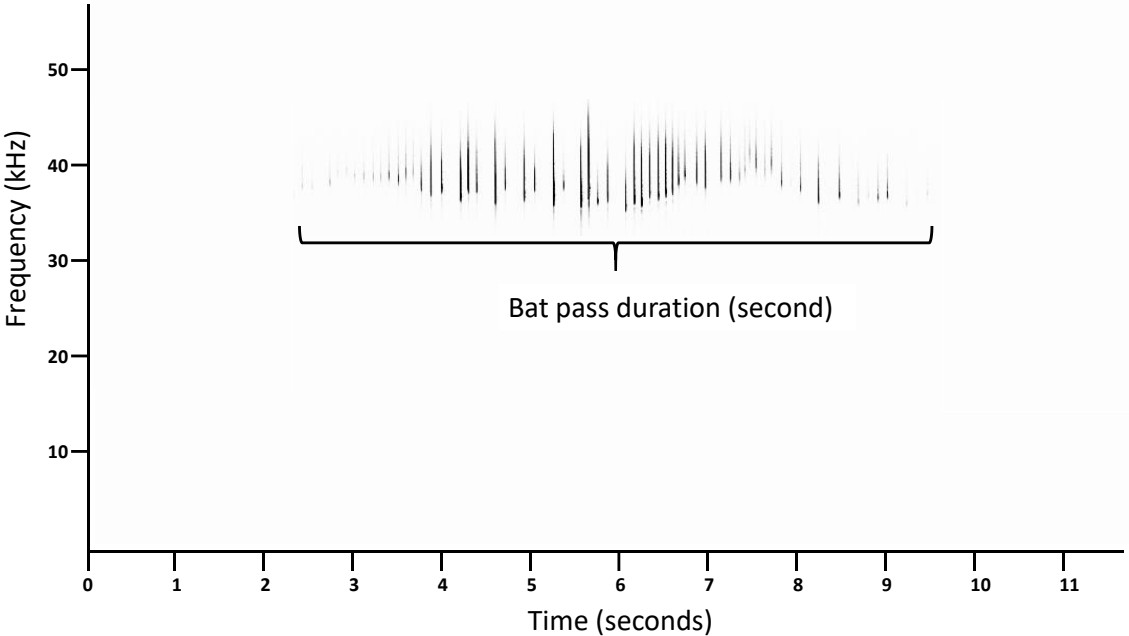

**Figure 1.** Bat-pass duration (*Pipistrellus kuhlii*, 512 Fast Fourier Transform size).

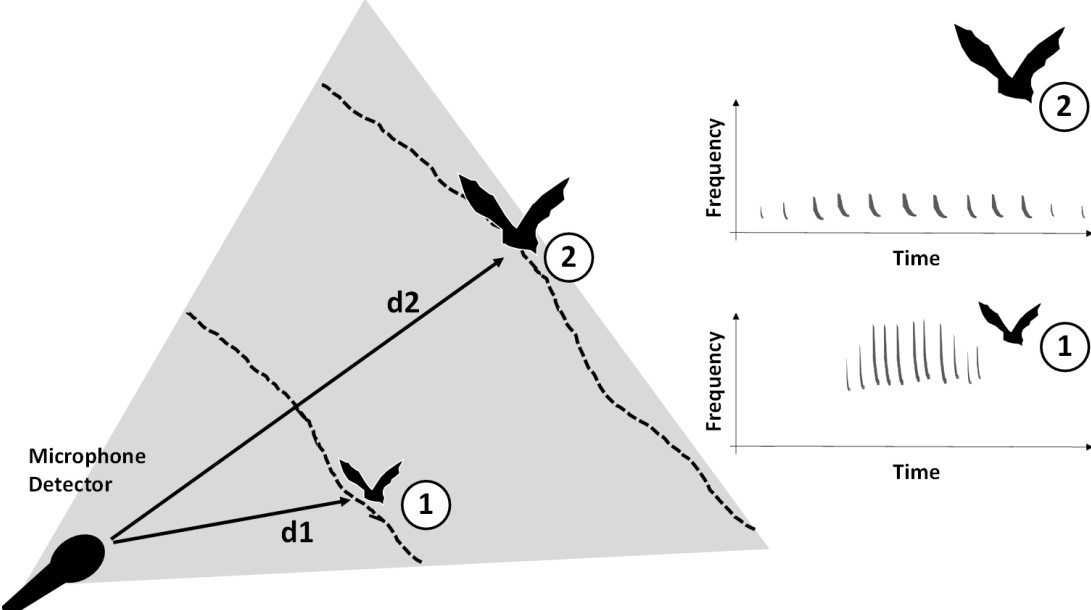

**Figure 2.** Link between bat trajectory and recording bat-passes for two species with different maximal distances of detection (d1, e.g., *a Myotis* spp.). and (d2, e.g., a *Nyctalus* spp.).

## 2. Materials and Methods

### 2.1. Bat Sampling

Data were provided by the French bat monitoring program (FBMP) [50], a citizen-science program that has been running since 2006 and is coordinated by the French Museum of Natural History (MNHN). The FBMP is based on standardized echolocation recordings [51]. Two different versions are used in the FBMP: a road survey by car and a point count. In both surveys, two detector models are used by the volunteer network: Tranquility Transect (a bat detector with a capacitive microphone designed by Courtpan Design Ltd., Cheltenham, UK) and D240x (a bat detector with an electret microphone designed by Pettersson Elektronik AB, Uppsala, Sweden). Bat calls are digitally recorded and stored on a secure digital card in Waveform Audio File Format (.WAV). Each site was monitored twice, first between 15 June and 31 July (during late pregnancy and lactation) and then between 15 August and 31 September (when weaned young are flying, and individuals are expected to be less dependent on their reproductive roost). The observers began their sampling as soon as 30 min after sunset, which varies from season to season and from year to year. Thus, this sampling overlays the peak activity of hawking bat species that begins 30 min after sunset and spans less than 3 h [52]. The observers sampled bats only when weather conditions were favorable (i.e., no rain, temperatures higher than 12 °C, 30 min after sunset, and without strong wind (<20 km/h)). Thus, the conditions when these data were recorded are close to conditions often selected for comparative studies [23,53].

The volunteers involved in the car-transect surveys recorded bat activity while driving at a constant low speed ($25 \pm 5$ km/h) along a route of at least 30 km within a 10-km radius around the volunteer's residence. Within this route, 10 random 2-km transects were recorded. Currently, the database is composed of 160 routes representing 1618 different 2-km transects (Table S2).

For the point-count survey, a 2 km$^2$ square was randomly chosen by the Museum within a radius of 10 km from the observer's home. Within the 2-km$^2$ square, a minimum of 10 points were sampled. The points in each square were sampled (i.e., continuously recorded for 6 minutes/point) during the same night. Currently, data have been gathered from 120 squares representing 1272 different recording points (Table S2).

## 2.2. Species Identification and Measurement of Bat-Pass Duration

Volunteers conducted the species acoustic identification, whereas the final data validation was conducted by museum experts. Some calls of *Myotis* (52%) were pooled into a *Myotis* spp. group due to identification uncertainties. Similarly, all calls from the *Plecotus* genus were pooled into a single group. Duration of the bat-passes in seconds was defined as the interval between the beginning of the first echolocation pulse detected to the end of the last pulse detected in a series emitted by an individual (Figure 1). Durations were measured using a cursor on the real-time spectrograms of an echolocation group of calls, using Syrinx software version 2.6 (Seattle, WA, USA) [54].

## 2.3. Statistical Analysis

We applied a generalized linear model (GLM) with a Poisson error distribution with the aim of evaluating how bat-pass duration varies among species relative to other variables, such as expecting bat-pass duration; temperature; humidity; the microphone of the detector (i.e., D240x or Tranquility Transect [46]); methods to survey bat activity (i.e., line transects and stationary measurement [51]); habitat; and volunteers who manually perform the measure. Habitat is a continuous index of clutter of the habitat (i.e., an explicit seven-class gradient of habitat structure, ranging from (1) open habitat, which is farmland and open fields without any trees or bushes, to (7), which is cluttered habitat provided by the FBMP. Following a multi-model inference [55], we generated a set of candidate models containing all possible variable combinations and ranked them using corrected Akaike information criterion (AICc) using the dredge function (R package MuMIn, Barton 2018). We only integrated the models that complied with the following conditions: (1) models do not simultaneously include correlated covariates ($R^2 > 0.7$) and (2) models do not include more than five variables to avoid over-parameterization. This resulted in a total model set of 79 models, with one model performing notably better than the others (Table S3)

To evaluate whether the measured bat-pass duration could be an acceptable proxy for the distance of detection for a particular species, tests were run to determine the association between the paired samples (Pearson correlation coefficient). We used the duration of the bat-passes provided by the two surveys along with the detection distances published by Barataud [3] of the corresponding species identified in the recording. Similar correlations were examined between Barataud's [3] distances of detection and call duration or frequency of peak energy from the study by Obrist et al. [4]. In addition, a generalized linear model (GLM with a Poisson error distribution) for the taxa with sufficient data was used to test the influence of detector type (Tranquility Transect vs. D240x) and survey protocol (car-transect survey vs. point-count survey) on bat-pass duration (response variable). A model was created for each species. The results were evaluated using a type-II ANOVA with an *F*-test. All analyses were performed with R statistical software (R Development Core Team. 2016 Vienna, Austria)

## 3. Results

From the FMBP dataset, we measured 25,786 bat-passes from 20 taxa (18 species and two genera, Table 1). In our dataset, bat species was one of the best predictors of bat-pass duration (41.8% of explained variance), followed by operator (29.4%), type of survey (27.2%), temperature (1.1%), and humidity (0.4%).

Strong relationships were detected between measured bat-pass duration and published detection distance for both survey methods (car-transect survey: $r = 0.929$, $p < 0.001$, Figure 3a; point-count survey: $r = 0.904$, $p < 0.001$, Figure 3b). A consistently strong correlation was also found between the bat-pass durations measured from the car-transect survey and those from the point-count survey ($r = 0.988$, $p < 0.001$). When significant, the longer bat-pass durations were measured by the point-count survey and the D240x detector (Table 2). A significant correlation was also found between frequency of peak energy and detection distance ($r = -0.60$, $p = 0.002$), whereas no correlation was found between call duration and detection distance ($r = 0.38$, $p = 0.071$). For the latter, the weak correlation was

obviously due to Rhinilophidae species (*Rhinolophus hipposideros* and *R. ferrumequinum*); when they are excluded from the analysis, the correlation is significant ($r = 0.921$, $p < 0.001$), Figure S1.

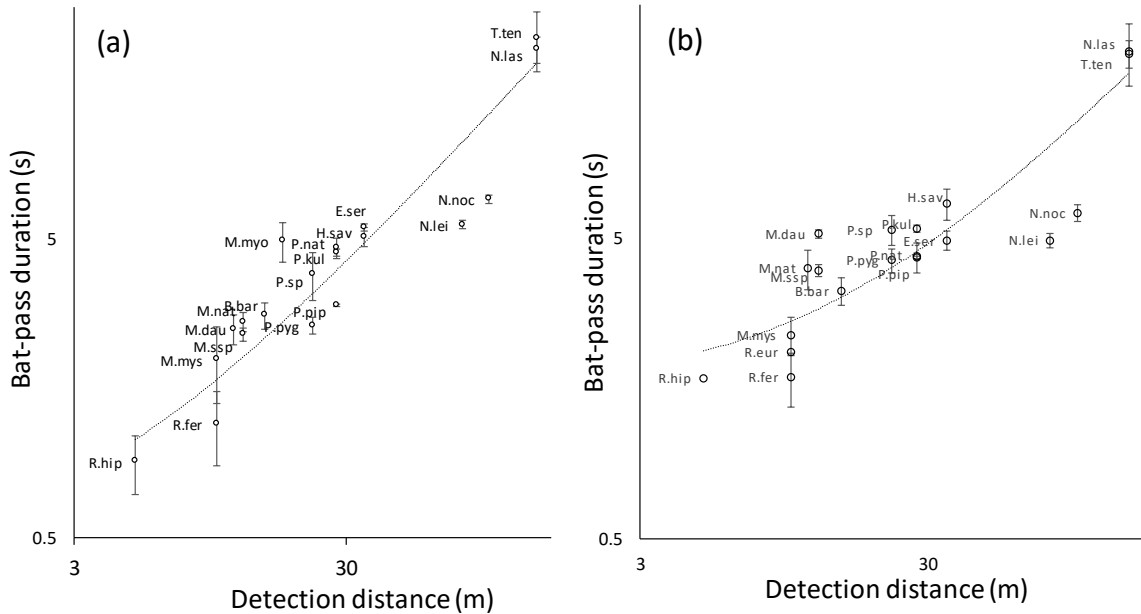

**Figure 3.** Relationship between bat-pass duration and detection distance [3] for (**a**) the car-transect survey (**b**) and the point-count survey. For species abbreviations, see Table 1.

**Table 1.** Means of bat-pass duration (BPD) expressed in seconds ± SE and number of bats call recorded (N) according to type of survey.

| Species | Abbreviations | Point-Count Survey | | Car-Transect Survey | |
|---|---|---|---|---|---|
| | | Mean BPD | N | Mean BDP | N |
| *Barbastella barbastellus* (Schreber, 1774) | (B.bar) | 3.342 ± 0.343 | 45 | 2.788 ± 0.263 | 132 |
| *Hypsugo savii* (Bonaparte, 1837) | (H.sav) | 6.525 ± 0.755 | 16 | 5.114 ± 0.366 | 182 |
| *Pipistrellus kuhlii* (Kuhl, 1817) | (P.kuh) | 5.397 ± 0.151 | 805 | 4.523 ± 0.112 | 981 |
| *Pipistrellus nathusii* (Keyserling & Blasius, 1839) | (P.nat) | 4.346 ± 0.490 | 72 | 4.695 ± 0.365 | 66 |
| *Pipistrellus pipistrellus* (Schreber, 1774) | (P.pip) | 4.340 ± 0.042 | 8116 | 3.013 ± 0.025 | 11,132 |
| *Pipistrellus_pygmaeus* (Leach, 1825) | (P.pyg) | 4.228 ± 0.400 | 92 | 2.577 ± 0.162 | 260 |
| *Eptesicus serotinus* (Schreber, 1774) | (E.ser) | 4.921 ± 0.359 | 130 | 5.502 ± 0.145 | 713 |
| *Nyctalus lasiopterus* (Schreber, 1780) | (N.las) | 21.013 ± 4.841 | 8 | 21.683 ± 2.189 | 15 |
| *Nyctalus leisleri* (Kuhl, 1817) | (N.lei) | 4.924 ± 0.284 | 226 | 5.606 ± 0.168 | 531 |
| *Nyctalus noctula* (Schreber, 1774) | (N.noc) | 6.093 ± 0.411 | 143 | 6.832 ± 0.221 | 414 |
| *Myotis* ssp. | (M.spp) | 3.905 ± 0.183 | 493 | 2.401 ± 0.123 | 274 |
| *Myotis daubentoni* (Kuhl, 1817) | (M.dau) | 5.178 ± 0.163 | 604 | 2.654 ± 0.199 | 54 |
| *Myotis mystacinus* (Kuhl, 1817) | (M.mys) | 2.387 ± 0.349 | 15 | 1.978 ± 0.573 | 9 |
| *Myotis myotis* (Borkhausen, 1797) | (M.myo) | - | 0 | 4.943 ± 0.761 | 28 |
| *Myotis nattereri* (Kuhl, 1817) | (M.nat) | 3.989 ± 0.599 | 27 | 2.503 ± 0.285 | 31 |
| *Plecotus* ssp. | (P.spp) | 5.340 ± 0.606 | 35 | 3.830 ± 0.651 | 57 |
| *Tadarida teniotis* (Rafinesque, 1814) | (T.ten) | 20.970 ± 2.134 | 44 | 23.564 ± 5.350 | 11 |
| *Rhinolophus euryale* (Blasius, 1853) | (R.eur) | 2.100 ± 0.280 | 2 | - | 0 |
| *Rhinolophus ferrumequinum* (Schreber, 1774) | (R.fer) | 1.725 ± 0.354 | 8 | 1.206 ± 0.337 | 9 |
| *Rhinolophus hipposideros* (Bechstein, 1800) | (R.hip) | 1.717 ± 0.224 | 3 | 0.900 ± 0.200 | 3 |

**Table 2.** Effect of survey type—car transect survey (CTS) vs. point-count survey (PCS)- and detector type (Tranquility Transect TT vs. D240x).

| Species | Survey-Type Effect | | Detector-Type Effect | |
|---|---|---|---|---|
| | Effect | *p*-Value | Effect | *p*-Value |
| *Barbastella barbastellus* (Schreber, 1774) | - | $p = 0.07$ | - | $p = 0.97$ |
| *Hypsugo savi* (Bonaparte, 1837) | - | $p = 0.06$ | - | $p = 0.13$ |

**Table 2.** *Cont.*

| Species | Survey-Type Effect | | Detector-Type Effect | |
|---|---|---|---|---|
| | Effect | *p*-Value | Effect | *p*-Value |
| *Pipistrellus kuhlii* (Kuhl, 1817) | CTS < PCS | $p = 0.001$ | TT < D240x | $p = 0.002$ |
| *Pipistrellus nathusii* (Keyserling & Blasius, 1839) | - | $p = 0.11$ | - | $p = 0.07$ |
| *Pipistrellus pipistrellus* (Schreber, 1774) | CTS < PCS | $p < 0.001$ | TT < D240x | $p < 0.001$ |
| *Pipistrellus_pygmaeus* (Leach, 1825) | CTS < PCS | $p < 0.001$ | - | $p = 0.14$ |
| *Eptesicus serotinus* (Schreber, 1774) | - | $p = 0.21$ | - | $p = 0.05$ |
| *Nyctalus leisleri* (Kuhl, 1817) | - | $p = 0.05$ | - | $p = 0.44$ |
| *Nyctalus noctula* (Schreber, 1774) | - | $p = 0.16$ | - | $p = 0.52$ |
| *Myotis* ssp. | CTS < PCS | $p < 0.001$ | TT < D240x | $p < 0.001$ |
| *Myotis daubentoni* (Kuhl, 1817) | CTS < PCS | $p < 0.001$ | TT < D240x | $p < 0.001$ |

## 4. Discussion

The strong correlation between bat-pass duration and the known detectability distance for bat echolocation calls allowed us to consider bat-pass duration as a good proxy for bat detectability. Huge variations around the mean (Figure 3) suggest that bat-pass duration depends on several factors (species, operator, detector type, survey type, weather conditions), yet our dataset suggests that species identity is the best predictor. The strong correlation between bat-pass duration and the known detectability distance is stronger than other alternative call parameters, such as frequency of peak energy or call duration. For call duration, the weak correlation is due to *Rhinolophus hipposideros* and *R. ferrumequinum*. When these two species are excluded from the analysis, the correlation is significant. This is likely due to important differences in mechanisms involved in echolocation modalities within Rhinolophidae [56]. Certain species deviated slightly from the regression, regardless of the survey considered, such as *N. leisleri* (Kuhl, 1817) and *N. noctula* (Schreber, 1774). According to the measured bat-pass duration and the linear regression (Figure 2), we expected a detection distance of approximately 32 or 37 m for *N. leisleri* from the car-transect survey or the point-count survey, respectively, instead of the 80 m proposed by Barataud [3]. For *N. noctula*, we expected 43 or 47 m instead of the 100 m proposed by Barataud [3]. Anomalies in this linear relationship could be due to a difference in flight behavior among species. We expected that species such as *N. leisleri* and *N. noctula*, which fly much higher and exhibit relatively fast and straight trajectories, would produce shorter bat-pass durations on average than bats exhibiting more curved trajectories, such as *Myotis* [57].

The strong correlation found between estimations of distance of detection reported by Barataud [3] and those obtained with the car-transect survey or the point-count survey should not hide the fact that our correlations are based on one external data set, for which the methodology is poorly documented, and partially based on the experience and judgment of the author. It would be prudent to assume that Barataud's estimations [3] constitute a relative scale of distance of detection among species rather than accurate absolute distances of detection. This does not invalidate the overall findings because they are based on correlation tests. Note that the estimations of distance of detection published by Barataud [3] are currently the only ones available for the Western Palearctic, highlighting the need for additional research. Among the 11 species tested, when a bat detector effect was identified, the more directional detector (Tranquility Transect) recorded a shorter bat-pass duration, as expected. When a survey-type effect was identified, the bat-pass duration was greater with the point-count protocol than the car-survey transect, which was completed at a constant speed of 25 km/h. As expected, bat-pass duration and, in turn, detectability are partially dependent on the survey methods and are influenced by the choice of detector, and more specifically by its sensitivity and directionality, as suggested in previous studies [33,45,58,59]. Sensitivity and directionality are partially dependent: a detector with greater directionality sometimes exhibits a greater sensitivity due to a narrow and more elongated detection area. Thus, providing an accurate absolute distance of detection regardless of the technical choice appears to be unrealistic.

Theoretical detection distances could also be calculated based on the measurement of sound pressures of call emissions, spreading loss, and atmospheric transmission functions [32]; however, this approach needs the microphone to be highly precisely calibrated, so it is better suited for laboratory experiments than field experiments.

The advantages of bat-pass duration measures are that for each study with unique characteristics (choices of detectors and protocol design, habitats, and weathers conditions) experimenters can easily produce their own proxy for the distance of bat call detection, thus resulting in greater transparency, and therefore greater reproducibility, of the methodology. However, this statistical-based approach requires a considerable amount of data. Reducing variability in bat-pass duration should be promoted by using data from one type of survey and improving inter-observer reliability. Note, however, that despite having provided two days of training to our volunteers, and having employed software and common software configurations, the operator was still an important factor: 29.4% of the variability in bat-pass duration was attributable to the operator. Fortunately, the recently reduced cost of acoustic recorders has resulted in the development of automated, remotely deployed passive acoustic monitoring (PAM) systems. A PAM system produces a considerable amount of data because it can record overnight. Therefore, several hundreds or thousands of bat-passes can be regularly recorded each night at several sites, for multiple species [14,26], and even in less suitable habitats such as intensively managed crop fields [26]. In parallel to the development of PAM systems, software programs detecting sound events, extracting numerous features, and automatically identifying species have been developed [8] and may potentially contribute to the reduction of inter-observer variations. With that said, such automated identification software programs have been criticized due to significant error rates, suggesting cautious and limited use [60]. In response to these challenges, Barré [61] proposed a cautious method to account for errors in acoustic identifications without excessive manual checking of recordings, thereby helping to potentially unlock new and broader perspectives.

The proposed approach simply and indirectly measures detectability and could be used to monitor the degradation of a microphone (i.e., a loss of sensitivity of the bat detectors) in real time. Bat-pass duration for a species should not decrease significantly among years unless the microphones used have declined in sensitivity. Therefore, we recommend that monitoring projects intensively using bat detectors in the field, particularly for monitoring surveys to determine population trends [62], should test their microphones regularly, replace microphones with declining sensitivity, and record sensitivity with the aim to include sensitivity as a potential covariate in the statistical analyses of acoustic data. The monitoring of the degradation of the equipment could be completed in the laboratory, but it is not always possible during expeditions or within a citizen-science network where detectors circulate among volunteers. Experimental tests of the real effect of the degradation of the microphone on bat-pass duration still need to be completed.

Finally, the possibility of producing a detectability index allows for the creation of weighted activities measures for each species when species are pooled in the same index, such as the widely used "total activity". Without any correction, such indexes are built by adding species with very different distances of detection. For example, some *Rhinolophus* can be detected at distances <10 m, whereas some noctules can be detected at distances >150 m, which implies that the abundances of species that are less detectable are underweighted, whereas the abundances of highly detectable species are overestimated in the assessment of community metrics using abundance as a parameter (i.e., total activity and mean community trait). However, before using this detectability index for accurate weighting of a community's index, more studies are needed on species-specific patterns of the propagation of omnidirectional bats calls.

**Supplementary Materials:** The following are available online at http://www.mdpi.com/1424-2818/11/3/47/s1, Figure S1: Relationship between detection distance (Barataud's study [3]) and frequency of peak energy (Obrist et al.'s study [4]) and call duration (Obrist et al.'s study [4]); Table S1: Distance of detection from Barataud's study [3], frequency of peak and call duration energy from Obrist et al.'s study [4]; Table S2: Additional information on the French National Bat Monitoring Program coordinated by the National Museum of Natural History (MNHN); Table S3: Model set with df, log-likelihood, AICc, ΔAICc, and Akaike weights.

**Author Contributions:** C.K., Y.B., I.L.V., R.L., J.M., and J.-F.J. conceived and designed the experiments; C.K., Y.B., R.L., J.M., and J.-F.J. provided the data; C.K. analyzed the data; C.K. wrote the first draft version; C.K., Y.B., I.L.V., R.L., and J.M. reviewed and edited the final version.

**Funding:** This research received no external funding.

**Acknowledgments:** We express our deep gratitude to the Vigie-Nature volunteers for data collection, especially experienced volunteers involved in bat-pass duration measures or who provided specific sound files for uncommon species: Quentin Amand, Avana Andriamboavonjy, Ronan Arhuro, Mathilde Baradat, Flore Cambon, Julien Cavallo, Robin Derozier, Nicolas Fillol, Gregory Fiquet, Vincent Gibaud, Pascal Guichard, Emmanuel Jacob, Robin Julien, Marine Lauere, Lara Million, Julie Maratrat, Marion Parisot-Laprun, Laura Plichard, Loic Robert, Magali Roche Loic Salaun, Jean Claude Vandevelde, Arthur Vernet, and Sophie Wrobel. We also thank Jeanette Thomas and two anonymous reviewers for helpful comments and suggestions.

**Conflicts of Interest:** The authors declare no conflict of interest.

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
