# Peer review of "Bat Pass Duration Measurement: An Indirect Measure of Distance of Detection"

_diversity, doi:10.3390/d11030047_

Round 1
Reviewer 1 Report
This is an interesting short study with a neat logical argument leading to remarkable and simple results that are important for bat researchers using acoustic sampling methods. The methods cannot be faulted and the conclusions that are drawn are sound.However, I am not quite sure what is the take home message. This should be emphasied.
More importantly,
are the results of practical relevance? I am afraid that although the
results are statistically highly significant over this large dataset,
they would be of limited utility in bat studies.
Absolute
distances were not derived here; thus, densities cannot be computed
and to obtain them, reference measurements of detection distances of
each bat species should be made in each study. In studies where
different habitats are compared, detection distances would vary
between habitats and need to be measured first in each habitat. These
shortcomings or rather, the scope of the conclusions, should be
highlighted.
Moreover, I think the strong correlation was only obtained because of the averaging over such a large data set. Presumably, assuming bats fly more or less randomly, the correlation for much smaller, real-world data sets between single bat passes and detection distances would become much weaker. In fact, the actual detection distances are constantly changing because of the movement of the bats, and only the maximal distance distance would correlate with the average pass duration over a large aggregated data set. Indeed, the variation in bat pass durations as shown in Fig 2 are huge (the scale is logarithmic).
detailed comments:
Title: the title would be more informative if you removed the question mark
Introduction:
The first part describing the history and types of methods is quite lengthy.
L74: There are very few estimates of detection distances in sound recordings for any animals. We recently showed detection distances can be estimated for birds (Darras et al. 2018 MEE). But the challenge with the bats lies in the impossibility to intuitively estimate distances (we cannot hear bats and thus have no experience) and their constantly changing positions, I think this is worthy of mention. The approach we are using for measuring ultrasound detection spaces for bats is a simplified protocol derived from Darras et al. 2016 Biol Cons, simply measuring detection ranges by searching the distance at which an ultrasonic emitter is not recorded anymore by the recorder.
A conceptual figure with bat flight paths and radii of detection distances would show why long pass durations should correlate with them and be helpful to graphically and intuitively explain the concept (and also become a refreshing figure).
L40: “have matured”
L54: “calls” and maybe “sound recording”?
L57: delete “e”
L61: “an index”, “metric”
L63: Here the important point finally comes. maybe simply say “the environment” or the habitat
L70: what is meant with “bat transmitter”?
L79: “differences”
L83: maybe “turned on”?
L85: “individuals”
L94: “defining”
L95: “estimating”
L96-7: you should emphasize more clearly that any comparison of bat activity between two sites is biased if the sampling volumes are unknown (and they do differ between habitats).
L106: “using”
L107: integrate citation better into sentence. It would be good to shortly explain how such distances were obtained since it was stressed how difficult it is to obtain them before.
L109: “compared”, “metrics”
L113: “frequency has been”
L115: “and thus have a greater expected”
L116: “is”
L116-7: I don’t see why call duration should correlate with greater detection distances. Shortly mention?
Fig.1: “frequency”, “seconds” Which species is it? Give details of spectrogram (FFT, etc.)
Methods
I stopped checking English errors here. Please check the manuscript.
The detection distances from Barataud are central to this manuscript, yet details are missing. At the very least, they should have been obtained with the same microphones as the FBMP (otherwise only relative distances can be inferred) and also in comparable habitats (those are not specified either for the FBMP).
http://vigienature.mnhn.fr/page/vigie-chiro: link does not work
L140-150: having mentioned that detection distances depend on habitat earlier (and they do), it should be specified where the FBMP surveys were made
L158: why “train”?
L181: intuitively, that was to be expected
Figure 2: I think it is not necessary to list species’ taxonomic details here. Maybe it would be easier to add a column to Table 1 and refer to that. More labels for the axes are required.
L208-11: this is the second time I read this explanation, but I don’t see what is obvious here (for non-chiropteroligists). The second sentence is also vague.
L218-20: I was also thinking that flight speed would matter. If you have the data, it would be good to include it into the model.
L230: this is likely due to low microphone signal to noise ratio, see Darras et al. JAPPL 2018. Since you analyse the differences between recorder types statistically, it would be good to develop that technical topic a bit more.
L237-8: indeed, detection distances are dependent on the recording system
L239-42: a new method based on sound pressure levels (I don’t know the cited one) for birds should be published soon (it is in review) and it bears potential, especially for bats that are moving all the time and already detected automatically.
L251-6: recording calibrated ultrasounds would probably be a much more practical, accurate and direct measure of microphone sensitivity.
L263-6: this is a very important point worthy of development, it could be mentioned in the introduction as a justification for your study.
Thank you for your manuscript. I wanted to review it because the proposed idea that bat pass duration correlates with maximal detection range was surprising, but I was rapidly convinced by the clever finding.
Author Response
We thank Reviewer 1 for their relevant and useful comments. Responses to his main concerns are provided below. We have modified our text accordingly and explained the elements that were not clear in the first version of manuscript.
REVIEWER 1
This is an interesting short study with a neat logical argument leading to remarkable and simple results that are important for bat researchers using acoustic sampling methods. The methods cannot be faulted and the conclusions that are drawn are sound.
However, I am not quite sure what is the take home message. This should be emphasied.
More importantly, are the results of practical relevance? I am afraid that although the results are statistically highly significant over this large dataset, they would be of limited utility in bat studies.
Absolute distances were not derived
We we agree we didnot derived absolute distance, however relative distance of detection could be usefull for weighting activity measure for each species when they are pooled in the same index, such as the widely used “total activity.” Without any correction, such indexes are built by adding species with very different distance of dectection. This implies that the abundances of species that are less detectable are underweighted, while the abundances of highly detectable species are overestimated in the assessment of community metrics using abundance as a parameter (i.e., total activity and mean community trait).
This has been discussed lines 281-290
“Finally, the possibility of producing a detectability index allows the creation of weighted activities measures for each species when species are pooled in the same index, such as the widely used “total activity”. Without any correction, such indexes are built by adding species with very different distances of detection. For example, some Rhinolophus can be detected at distances <10 m, whereas some noctules can be detected at distances >150 m, which implies that the abundances of species that are less detectable are underweighted, whereas the abundances of highly detectable species are overestimated in the assessment of community metrics using abundance as a parameter (i.e., total activity and mean community trait). However, before using this detectability index for accurate weighting of a community’s index, more studies are needed on species-specific patterns of the propagation of omnidirectional bats calls.”
here; thus, densities cannot be computed and to obtain them, reference measurements of detection distances of each bat species should be made in each study. In studies where different habitats are compared, detection distances would vary between habitats and need to be measured first in each habitat. These shortcomings or rather, the scope of the conclusions, should be highlighted.
Yes, we agree that detection distances would vary between habitats and that acoustic recordings are expected to be influenced by many other factor. According to reviewer comments, we thus perform additional analysis with the aim to evaluate the relative importance of bat species compare to other variables such as operator (i.e. volunteers who perform manually the measure), temperature, humidity, the microphone of the detector (i.e. D240X or Tranquility transect ), methods to survey bat activity (i.e., line transects and stationary measurement) and an index of clutter of the habitat. Results show that several variable influence bat-pass duration but our results highlight that species was the best predictor of bat-pass duration.
We add lines 166-173
“We applied a generalized linear model (GLM) with a Poisson error distribution with the aim of evaluating how bat-pass duration varies among species relative to other variables, expecting bat pass duration: temperature; humidity; the microphone of the detector (i.e., D240x or Tranquility Transect [56]); methods to survey bat activity (i.e., line transects and stationary measurement [57]); habitat; and volunteers who manually perform the measure. Habitat is a continuous index of clutter of the habitat (i.e., an explicit seven-class gradient of habitat structure, ranging from (1) open habitat, which is farmland and open fields without any trees or bushes, to (7), which is cluttered habitat provided by the FBMP.”
We add lines 193-196
“From the FMBP dataset, we measured 25,786 bat passes from 20 taxa (18 species and two genera, Table 1). In our dataset, bat species was one of the best predictors of bat-pass duration (41.8% of explained variance), followed by operator (29.4%), type of survey (27.2%), temperature (1.1%), and humidity (0.4%).”
We add supplementary material
Table S3. Model set with df, log-likelihood, AICc, ΔAICc and ‘Akaike weights. Abbreviations: methods to survey bat activity (Surv.), operator (Op.), species (Sp.), humidity (Hum.), temperature (Temp.), index of clutter (Ind.Clutt.).
Model | df | log-likelihood | AICc | delta | weight |
Surv. + Op. + Sp + Hum + Temp. | 50 | -338618.1 | 677336.4 | 0.00 | 1 |
Surv. + Op. + Sp + Ind.Clutt + Temp. | 50 | -338654.9 | 677410.0 | 73.65 | 0 |
Null | 1 | -376925.1 | 753852.2 | 76515.84 | 0 |
In addition The advantages of bat-pass duration measures lie in the fact that for each study with unique characteristics (choices of detectors and protocol design, habitat, weathers condition), experimenters can easily produce their own proxy for the distance of bat call detection with transparency in the methodology used and thus reproducibility.
This has been discussed lines 261-264
“The advantages of bat-pass duration measures is that for each study with unique characteristics (choices of detectors and protocol design, habitats, and weathers conditions), experimenters can easily produce their own proxy for the distance of bat call detection with transparency, and thus reproducibility, of the methodology.”
Moreover, I think the strong correlation was only obtained because of the averaging over such a large data set. Presumably, assuming bats fly more or less randomly, the correlation for much smaller, real-world data sets between single bat passes and detection distances would become much weaker.
We agree with reviewer comment, (i) with smaller dataset, statistical powerful to detect correlations will decrease and (ii) accurate estimates of duration of an echolocating-bat pass for estimate proxy of detection distance will loss of interest
However we do not share the reviewer opinion, about the difference between our data set size (i.e. 25786 bat passes) and those from the “real-world data sets“ size. Because with the recently reduced cost of acoustic recorders has resulted in the development of automated, remotely deployed Passive Acoustic Monitoring (PAM) systems. PAM produces a considerable amount of data because it can record overnight. Therefore, like in our study several hundreds or thousands of bat passes can be regularly recorded each night, at several sites, for multiple species
For exemple see
Charbonnier et al. 2014. recorded “49271 passes identifiable to species or sonotype level. They include 27997 (56.9% of the identifiable passes) P. kuhlii passes, 17300 (35.1%) E. serotinus - N. leisleri sonotype passes, and 3015 (6.1%) P. pipistrellus passes. The other 1.8% were assigned to Myotis spp (341 passes), Plecotus spp (299), Nyctalus noctula (109), Barbastella barbastellus (94), Nyctalus lasiopterus (13) and Rhinolophus ferrumequinum (1)”
Millon et al., « recorded,9900 bat passes"
Put et al. 2018 “recorded 15,430 bat passes. All seven species present in the study area were represented: big brown bat (2642 passes), eastern red bat (141), hoary bat (11,368), little brown bat (513), northern long-eared bat (35), silver-haired bat (144) and tri-colored bat (93)
Barré et al. 2018 “reccorded 193,980 bat passes of 8 species and 3 species groups were recorded at the 207 study sites, where the most abundant species was P.pipistrellus, representing 81% of the observations. The least abundant species were R. ferrumequinum (22 bat passes) and N. noctula (25 bat passes), which were present in 7 and 9% of the study sites, respectively.”
Claireau et al. 2019 recorded 57,941 bat passes
This information is included in the revised version, lines 265-269
“Fortunately, the recently reduced cost of acoustic recorders has resulted in the development of automated, remotely deployed Passive Acoustic Monitoring (PAM) systems. PAM produces a considerable amount of data because it can record overnight. Therefore, several hundreds or thousands of bat passes can be regularly recorded each night at several sites, for multiple species [14,26], and even in less suitable habitats such as intensively managed crop fields [26].”
And according to reviewer comment we highlight the need to sufficient amount of data for perform such estimate lines 264-265
“However, this statistical-based approach requires a considerable amount of data..”
In fact, the actual detection distances are constantly changing because of the movement of the bats, and only the maximal distance distance would correlate with the average pass duration over a large aggregated data set. Indeed, the variation in bat pass durations as shown in Fig 2 are huge (the scale is logarithmic).
We agree the distance of detection are constantly changing because of the movement of the bats, and depend of several factors (habitat, detector, weather conditions;..) with is illustrated by the huge variation among the mean however additional analysis suggest that species remain the main factor influencing bat-pass duration
We add lines 222-225
“Huge variations around the mean (Figure 3) suggest that bat-pass duration depends on several factors (species, operator, detector type, survey type, weather conditions), yet our dataset suggests that species identity is the best predictor.”
detailed comments:
Title: the title would be more informative if you removed the question mark
Done
Introduction:
The first part describing the history and types of methods is quite lengthy.
This seemed to us necessary and beneficial to the understanding and implications of this article, particularly for “bat activity” description, a metric uses in numerous bats studies but which include a wide spectrum of methodologies for the assessment of this metric
L74: There are very few estimates of detection distances in sound recordings for any animals. We recently showed detection distances can be estimated for birds (Darras et al. 2018 MEE). But the challenge with the bats lies in the impossibility to intuitively estimate distances (we cannot hear bats and thus have no experience) and their constantly changing positions, I think this is worthy of mention.
We agree and added lines 64-66
“One challenge with the bats is, in comparison with other taxa such as diurnal birds, difficulties in intuitively estimating distances..”
The approach we are using for measuring ultrasound detection spaces for bats is a simplified protocol derived from Darras et al. 2016 Biol Cons, simply measuring detection ranges by searching the distance at which an ultrasonic emitter is not recorded anymore by the recorder.
Yes, this is probably the method used by the study we cited (Barataud 2015), lines 76-79
“Without knowing the specific details of the methodology used by Barataud [3] to assess those distances, we assumed that the method consisted of the capturing and marking of bats using a chemiluminescent tag and visually estimating the distance from observer to bat in flight at night [38–40].”
A conceptual figure with bat flight paths and radii of detection distances would show why long pass durations should correlate with them and be helpful to graphically and intuitively explain the concept (and also become a refreshing figure).
According to reviewer suggestion, we include a figure 2 with the aim to illustrate the expect relationship between radii of detection distance and pass duration for two contrasted species.
L40: “have matured”
Checked
L54: “calls” and maybe “sound recording”?
Checked
L57: delete “e”
Checked
L61: “an index”, “metric”
Checked
L63: Here the important point finally comes. maybe simply say “the environment” or the habitat
Done
L70: what is meant with “bat transmitter”?
We change : “the transmitter (i.e. bats)
“
L79: “differences”
Checked
L83: maybe “turned on”?
Checked
L85: “individuals”
Checked
L94: “defining”
Checked
L95: “estimating”
Checked
L96-7: you should emphasize more clearly that any comparison of bat activity between two sites is biased if the sampling volumes are unknown (and they do differ between habitats).
Yes, we added lines 88-89
“Estimating the volume of airspace sampled is an ongoing issue because comparisons of bat activity between sites could be biased by the environment (e.g., habitats),”
L106: “using”
Checked
L107: integrate citation better into sentence. It would be good to shortly explain how such distances were obtained since it was stressed how difficult it is to obtain them before.
We include Barataud reference, so it make linked with the sentence that shortly explain how such distances were obtained.
“we assumed that the method consisted of the capturing and marking of bats using a chemiluminescent tag and visually estimating the distance from observer to bat in flight at night [38–40].”
L109: “compared”, “metrics”
Checked
L113: “frequency has been”
Checked
L115: “and thus have a greater expected”
Checked
L116: “is”
Checked
L116-7: I don’t see why call duration should correlate with greater detection distances. Shortly mention?
Call intensity is linked to distance of detection, and call duration is positively correlated to call intensity, thus we can expect call duration should correlate with greater detection distances
We added a Figure (Fig. 2) for illustrate this relation ship
We add lines 110-115:
“For many echolocating bats, the peak frequency has been shown to be negatively correlated to body size [42,49]. Larger bats produce lower frequency sounds because they have a bigger larynx and larger resonant chambers [50]. Yet, calls of larger species are more intense [42] and thus have an expected greater distance of detection (Figure 2). Call duration is positively correlated with call intensity [48], so species with long pulsation duration are expected to have a greater distance of detection (Figure 2).”
Fig.1: “frequency”, “seconds” Which species is it? Give details of spectrogram (FFT, etc.)
Checked
Methods
I stopped checking English errors here. Please check the manuscript.
We send the revised manuscript to MDPI editing service
The detection distances from Barataud are central to this manuscript, yet details are missing. At the very least, they should have been obtained with the same microphones as the FBMP (otherwise only relative distances can be inferred) and also in comparable habitats (those are not specified either for the FBMP).
The detection distances provides by Barataud are order of magnitude in 5meters intervals, without details about microphones used and habitat. We considered them as “relative distances” This is not a problem because we perform correlation between detection distances from Barataud (2015) and bat pass duration, so no absolute value are needed.
We added lines 81-82
“Barataud [3] provided no details about microphone used or habitats; thus, these empirical measures should be regarded rather as relative distances of detection.”
http://vigienature.mnhn.fr/page/vigie-chiro: link does not work
Sorry, the FBMP site has been upgrade so we update the link
http://www.vigienature.fr/fr/chauves-souris
L140-150: having mentioned that detection distances depend on habitat earlier (and they do), it should be specified where the FBMP surveys were made
All the main habitat present in France are survey and according to the FBMP sampling design, it result in a survey of habitats that are quite representative of French land cover (see Kerbiriou et al. 2018)
L158: why “train”?
We change by “group of calls”
L181: intuitively, that was to be expected
Figure 2: I think it is not necessary to list species’ taxonomic details here. Maybe it would be easier to add a column to Table 1 and refer to that. More labels for the axes are required.
According to reviewer comment we add a column in Table 1 for abbreviations
L208-11: this is the second time I read this explanation, but I don’t see what is obvious here (for non-chiropteroligists). The second sentence is also vague.
We are not sure I understood correctly, it seem to us that no implicit chiropterologists information is included in this sentence. Rhinolophidae species are clearly the species furthest from the regression see Supplementary information, and furthermore as mentioned in Result section, when this two species are excluded from the regression the Rsquare increase drastically from 0.38 to 0.92
L218-20: I was also thinking that flight speed would matter. If you have the data, it would be good to include it into the model.
Unfortunately, these data collected with only one microphone, thus we do not have such information
L230: this is likely due to low microphone signal to noise ratio, see Darras et al. JAPPL 2018. Since you analyse the differences between recorder types statistically, it would be good to develop that technical topic a bit more.
Yes, it could be due to low microphone signal to noise ratio or other intrinsic characteristics of the microphone, here we just highlight the need to take care about devices without focusing in detail on the intrinsic characteristic of each bat detector (response curve, gain…)
L237-8: indeed, detection distances are dependent on the recording system
Yes
L239-42: a new method based on sound pressure levels (I don’t know the cited one) for birds should be published soon (it is in review) and it bears potential, especially for bats that are moving all the time and already detected automatically.
We will expect the publication of this paper….
L251-6: recording calibrated ultrasounds would probably be a much more practical, accurate and direct measure of microphone sensitivity.
Yes, but within nationwide monitoring scheme involving hundreds of volunteers, calibration and regular monitoring of the degradation of the equipment in laboratory is not always realistic, thus our approach provide a pragmatic solution for detect a posteriori degradation of microphone.
L263-6: this is a very important point worthy of development, it could be mentioned in the introduction as a justification for your study.
We highlight this point in introduction, lines 91-96
“Assessing differences in detectability across bat species allows a more accurate activity measure, particularly when species are pooled in the same index, such as the widely used “total activity” or other community metrics within studying bat assemblages [35,46]. Without any correction of this index, the abundances of species that are less detectable are underweighted.”
Thank you for your manuscript. I wanted to review it because the proposed idea that bat pass duration correlates with maximal detection range was surprising, but I was rapidly convinced by the clever finding.
Thank you for your deep and helpful review
Reviewer 2 Report
I think this is an interesting study that has presented a method that could be helpful to standardize research on bats in the field. I think these results suggest that this method is worthy of further study, and so its publication would be valuable. There are issues with the ms that I think need to be addressed before it is ready for publication.
The first is a thorough examination of the text for grammar/language usage issues. In some cases these are minor distractions (e.g., on line 61, "indix" instead of "index") to issues that make the text difficult to understand interpret (e.g., the sentence starting on line 97 with "Assess differences..."). These definitely need to be addressed before the ms is suitable for publication.
From a methodological standpoint, I think the general technique is a good starting point (although more study is definitely needed to verify its reliability) but I did have some questions about the techniques used in the data collection. For example, the settings used for the time expansion on the detectors might have implications for the analysis of the recordings. While these recordings were not produced by the authors, they should still have attempted to present that information. Similarly, the settings on the digital recorder used to record the calls could be an issue (e.g., this particular device has settings like automatic gain control that would affect these results). It's important to know the background of the recordings as the analysis results depend on them. It seems that this information would be easy to obtain from the source of the recordings.
I also have a concern about the method used for measuring the duration of the passes as this is the vital piece in the experiment. Specifically, the use of a person manually measuring the duration of a pass on a computer using a cursor on a spectrogram is a potential problem.This can be affected by the settings for the spectrogram as well as the criteria for the individual doing the measuring. In addition, because the beginning and ending of a call pass would be the faintest, there is room for error in the measurements that might have a significant impact on the results. I think there at least needs to be a discussion of methods that were taken to ensure inter-observer reliability. I don't think re-doing all the measurements is going to be reasonable, but I think the ms needs to address those issues, indicate what was done to limit problems, etc.
Lastly, the concept of using this technique to measure loss of sensitivity in a microphone seems like it's placed in as an afterthought. It would be better to actually show how quickly degradation affected the measurement. It does seem that the degradation of the microphone would have an impact on these measurements, but many other factors would be able to impact the measurements as well. I think that this needs to have some data to show the impacts to be able to suggest that this sort of change is a reliable indicator of microphone degradation.
Author Response
REVIEWER 2
I think this is an interesting study that has presented a method that could be helpful to standardize research on bats in the field. I think these results suggest that this method is worthy of further study, and so its publication would be valuable. There are issues with the ms that I think need to be addressed before it is ready for publication.
The first is a thorough examination of the text for grammar/language usage issues. In some cases these are minor distractions (e.g., on line 61, "indix" instead of "index") to issues that make the text difficult to understand interpret (e.g., the sentence starting on line 97 with "Assess differences..."). These definitely need to be addressed before the ms is suitable for publication.
The manuscript was addressed to MDPI editing services
We rewrite sentence starting on line 97
“Assessing differences in detectability across bat species allows a more accurate activity measure, particularly when species are pooled in the same index, such as the widely used “total activity” or other community metrics within studying bat assemblages [35,46].”
From a methodological standpoint, I think the general technique is a good starting point (although more study is definitely needed to verify its reliability) but I did have some questions about the techniques used in the data collection. For example, the settings used for the time expansion on the detectors might have implications for the analysis of the recordings. While these recordings were not produced by the authors, they should still have attempted to present that information. Similarly, the settings on the digital recorder used to record the calls could be an issue (e.g., this particular device has settings like automatic gain control that would affect these results). It's important to know the background of the recordings as the analysis results depend on them. It seems that this information would be easy to obtain from the source of the recordings.
From the two possible outputs (the expansion and high frequency).of the detectors (Tranquility transect and D240x) we only used the high frequency output, so settings used for the time expansion (i.e. time expansion record length) did not interfere
We add supplementary details in Table S2: Additional information “Bat recording characteristics” about settings used, particularly for the trigger position
I also have a concern about the method used for measuring the duration of the passes as this is the vital piece in the experiment. Specifically, the use of a person manually measuring the duration of a pass on a computer using a cursor on a spectrogram is a potential problem. This can be affected by the settings for the spectrogram as well as the criteria for the individual doing the measuring. In addition, because the beginning and ending of a call pass would be the faintest, there is room for error in the measurements that might have a significant impact on the results. I think there at least needs to be a discussion of methods that were taken to ensure inter-observer reliability. I don't think re-doing all the measurements is going to be reasonable, but I think the ms needs to address those issues, indicate what was done to limit problems, etc.
We agree that use of a person manually measuring the duration of a pass on a computer may potentially generated variation among operator. In addition, many other factors (previous reviewer comment) may influence bat-pass duration. We thus perform additional analysis with the aim to evaluate the relative importance of bat species compare to other variables such as operator (i.e. volunteers who perform manually the measure), temperature, humidity, the microphone of the detector, methods to survey bat activity and habitat.
Results show that several variable influence bat-pass duration but our results highlight that species was the best predictor of bat-pass duration.
We add in Introduction section, lines 99-101
“We hypothesized that the measure of bat-pass duration (i.e., each event expressed in seconds) of a bat detected within the area of a bat detector (Figure 1): (1) vary among species and (2) is correlated with the distance of detection.”
We add in method section, lines 166-179
"We applied a generalized linear model (GLM) with a Poisson error distribution with the aim of evaluating how bat-pass duration varies among species relative to other variables, expecting bat pass duration: temperature; humidity; the microphone of the detector (i.e., D240x or Tranquility Transect [56]); methods to survey bat activity (i.e., line transects and stationary measurement [57]); habitat; and volunteers who manually perform the measure. Habitat is a continuous index of clutter of the habitat (i.e., an explicit seven-class gradient of habitat structure, ranging from (1) open habitat, which is farmland and open fields without any trees or bushes, to (7), which is cluttered habitat provided by the FBMP. Following a multi model inference [58], we generated a set of candidate models containing all possible variable combinations and ranked them using corrected Akaike information criterion (AICc) using the dredge function (R package MuMIn, Barton 2018). We only integrated the models that complied with the following conditions: (1) models do not simultaneously include correlated covariates (R² > 0.7) and (2) models do not include more than five variables to avoid over-parameterization. This resulted in a total model set of 79 models, with one model performing notably better than the others (Table S3)"
We add in Result section, lines 194-196
“In our dataset, bat species was one of the best predictors of bat-pass duration (41.8% of explained variance), followed by operator (29.4%), type of survey (27.2%), temperature (1.1%), and humidity (0.4%).”
We add in Discussion section, lines 222-225
“Huge variations around the mean (Figure 3) suggest that bat-pass duration depends on several factors (species, operator, detector type, survey type, weather conditions), yet our dataset suggests that species identity is the best predictor.”
We add a supplementarty material (Table S3) showing detail about model selection
Lastly, the concept of using this technique to measure loss of sensitivity in a microphone seems like it's placed in as an afterthought. It would be better to actually show how quickly degradation affected the measurement. It does seem that the degradation of the microphone would have an impact on these measurements, but many other factors would be able to impact the measurements as well. I think that this needs to have some data to show the impacts to be able to suggest that this sort of change is a reliable indicator of microphone degradation.
We agree with rewiever comment and add lines 279-280
“Experimental tests of the real effect of the degradation of the microphone on bat-pass duration still needs to be completed.”
Round 2
Reviewer 2 Report
I think the authors have done a good job addressing many of my comments. I do think a little more attention needs to be paid to the effect of the individual doing the measurements. The authors report that 29.4% of the variability in bat-pass duration was attributable to the operator and another 27.2% was attributed to the type of survey. Given that 41.8% was due to species it seems that reducing other sources of variation is still important. By only studying one type of survey, that variation can be removed, but the operator would still be an important factor. I would like to see some discussion of what methods were used in the measurements (which it seems would have to come from the museum that provided the recordings?) and a discussion of what could be done to reduce inter-operator variability. I don't think this needs to involve re-analysis of the current dataset, but future researchers would really benefit from having guidelines that would help them reduce this problem. I can't imagine it can be completely removed without a completely automated measuring system, which has problems of its own.
Author Response
I think the authors have done a good job addressing many of my comments. I do think a little more attention needs to be paid to the effect of the individual doing the measurements. The authors report that 29.4% of the variability in bat-pass duration was attributable to the operator and another 27.2% was attributed to the type of survey. Given that 41.8% was due to species it seems that reducing other sources of variation is still important. By only studying one type of survey, that variation can be removed, but the operator would still be an important factor. I would like to see some discussion of what methods were used in the measurements (which it seems would have to come from the museum that provided the recordings?) and a discussion of what could be done to reduce inter-operator variability. I don't think this needs to involve re-analysis of the current dataset, but future researchers would really benefit from having guidelines that would help them reduce this problem. I can't imagine it can be completely removed without a completely automated measuring system, which has problems of its own
According to reviewer comments, we disscuss the need to reduce variability in bat pass duration (lines 268-269)
"Reducing variability in bat-pass duration should be promoted: use data from one type of survey, and improve inter-observer reliability."
We recognized that operator would still an important factor and give details about the actions undertaken to ensure inter-observer reliability (line 269-271)
" Note, however, despite we provide to our volunteers 2 day training, a software and common software configurations, the operator still be an important factor: 29.4% of the variability in bat-pass duration was attributable to the operator."
We provide in Table S2 the weblink to the software configuration use by all volunteers
In addition we discuss the opportunities and limits to use automated measuring system
"In parallel to the development of PAM, software detecting sound events, extracting numerous features, and automatically identifying species have been developed [63] and may potentially contribute to reduce inter-observer variations. Such automated identification software have been criticized due to significant error rates, suggesting cautious and limited use [64]. In response to these challenges, Barré [65] propose a cautious method to account for errors in acoustic identifications without excessive manual checking of recordings, unlocking new and broader perspectives."